# Assessment of the Competitiveness of Islamic Fintech Implementation: A Composite Indicator for Cross-Country Analysis

**Sofya Glavina *** , **Irina Aidrus and Anna Trusova**

Institute of World Economy and Business, RUDN University, 117198 Moscow, Russia; aydrus-ia@rudn.ru (I.A.); hanna.trusova@yahoo.com (A.T.)
* Correspondence: glavina-sg@rudn.ru; Tel.: +7-(916)-925-47-06

**Abstract:** Islamic fintech is growing fast, especially in the Organisation of Islamic Cooperation (OOIC) member countries. In recent years, it has become one of the driving forces for the Islamic financial industry. Though the pandemic negatively affected global financial business, including conventional and Islamic segments, Islamic fintech has continued its steady development. i-Fintech increases access to Islamic financial services and financial inclusion in general to provide ESG-rich investment opportunities. The rise of Islamic fintech can help countries become financial hubs and promote sustainable development goals. This paper is aimed at designing an original composite indicator of the competitiveness of Islamic fintech adoption in order to perform a comprehensive assessment of the competitive advantages that are being used across various countries. The research methodology includes data for 65 countries where Islamic fintech companies are represented. We analysed 31 variables describing the development of Islamic financial technologies in each country and combined them into five categories included in the composite indicator. Key factors that determine the development of Islamic financial technologies in different countries around the globe are singled out. The economies with the highest scores are analysed to define their strengths and weaknesses. The practices of the leading countries that address identified vulnerabilities are described.

**Keywords:** Islamic finance; fintech; Islamic fintech; index; digitalisation

## 1. Introduction

Although the history of Islamic economic thought is as long as that of Islam itself, the Islamic finance industry in its modern form is relatively young. While conventional finance has been around for almost two centuries, the first experiments in establishing interest-free banks took place in 1963 in Egypt and Malaysia. Since then, Islamic finance has developed in terms of both products and infrastructure, providing options for those who seek services in line with their faith.

Today, the Islamic segment of the international financial system is still far less extensive than conventional finance. As of 2019, the industry's assets value was estimated at USD 2.88 trillion globally (State of the Global Islamic Economy Report 2020–2021). Nonetheless, the Islamic financial model has improved over time, functioning side by side with a ubiquitous and already extremely well-built conventional financial system. In order to provide Muslim consumers and entrepreneurs with economic opportunities equal to those derived from conventional finance, initiatives were launched to create financial instruments and institutions based on the principles of Islam. Thus, Islamic finance is a set of instruments, operating models, and financial institutions that comply with Sharia law.

Modern Islamic finance includes Islamic banking, Islamic insurance (takaful), Islamic funds, the sukuk market, and other financial institutions. The need for a separate segment of regulators that are specific to countries with an Islamic financial model follows necessarily from the very nature of Islamic finance. In addition to the Shariah supervisory boards

(SSBs) built right into the organisational structure of Islamic financial institutions, there are centralised Shariah committees at the national level in Muslim-majority countries. Moreover, international financial interaction and, hence, globalisation of Islamic finance required supranational regulators, among which, for example, are the Accounting and Auditing Organisation for Islamic Financial Institutions (AAOIFI) and the Islamic Financial Services Board (IFSB).

Despite the fact that Islamic finance is a relatively new sector of the global economy, the internal concept embedded in Islamic financial instruments draws ever-increasing attention every year. The 2008 global financial crisis is believed to be one of the Islamic financial model awareness drivers. At the time, Islamic banks showed themselves to be more resilient institutions with an internal safety margin enabled by the peculiarities of their operating model. Moreover, the financial and DEA analysis done by Musa, Naturin, Musova, and Durana claims that Islamic banks are more efficient than traditional ones (Musa et al. 2020). In addition, the principles of Islam, being the foundation of Islamic finance, are aligned naturally with the UN Sustainable Development Goals (SDGs); therefore, it is even more crucial to promote the understanding of the Islamic financial model globally.

As the Islamic financial industry develops and awareness of this model grows, the number of educational programs (professional training and academic degrees) in Islamic finance offered globally is increasing, including at world-leading universities, though the basic-level programs prevail (Asmyatullin 2020). Therefore, deeper and more specialised courses and programs in Islamic finance are required, particularly in Islamic fintech.

However, the current economic reality is threatened by a different kind of crisis; therefore, both conventional and Islamic financial institutions are equally endangered. It has been proven that in both banking systems an increase in new loans reduces the liquidity ratio (1 p.p. by 2p.p.) (Musa et al. 2021). Several Islamic banks reported losses or decreased profits in the second quarter of 2020 compared with the same period in the previous year (Zawya 2020). The main reason is a higher level of defaults on loans and decreasing quality of assets. The IFIs' activities are based on the demand from small and medium-sized businesses, their dependence being deeper than that of conventional FIs. Thus, the Islamic financial sector is supposed to suffer more losses this time. In this context, the approaches to the transformation of operating models in financial institutions are attracting much attention around the world. From this point of view, the pandemic is being recognised as a turning point for the digitalisation of the industry. Fintech utilisation was boosted everywhere; it facilitated consumption amid the fast-spreading crisis and reduced risks of contacts with other people (Vasenska et al. 2021).

Like other sectors, the Islamic financial industry has responded promptly to the demand from isolated clients, providing digital financial services of a large number and improved quality. Fintech adoption in Islamic finance is extremely important, though the response of Islamic financial institutions to the emergence of Fintech projects and its potential impact seems to be slower than that of their conventional counterparts (Ali et al. 2019).

Fintech's promise of enhanced effectiveness within the spectrum of IFIs' operations was noted by researchers long before the pandemic (Evans 2015; Hazik and Hassnian 2018). In addition to Islamic finance incumbents adapting to the demand for digitalisation, Sharia-compliant financial products and services today are being provided by other companies and digital platforms. Islamic Fintech expands opportunities for access to Sharia financial services and provides investment opportunities (Subagiyo 2019). In general, Sharia fintech increases financial inclusion.

The consequences of the pandemic seemed unpredictable for such a young segment as Islamic fintech (or "i-fintech" as it is referred to in Billah 2021). The new coronavirus posed challenges for the Islamic financial industry, and fintech assumed a special role in addressing them (Hassan et al. 2020). However, at the same time, it is suggested that Islamic finance in combination with fintech can help overcome the economic consequences of COVID-19 and reach out to the affected (Rabbani et al. 2021).

In the course of the year, the industry has seen noticeable growth in the number of representative companies, geographical diversification, and an expanded variety of services provided. To date, the total number of fintech companies that operate in accordance with Shariah rules has reached 278 globally (Fintech Landscape 2021). Additionally, there will be further growth in the number of i-fintech companies and the diversity of their activities and values. According to the Global Islamic Fintech Report (2021), the value of transactions will reach USD 128 billion by 2025 at a 21% compound annual growth rate, which is 6% higher than conventional fintech.

According to the IFN classification, a company may belong to one of the following verticals depending on the services provided and the technologies used:

(1)     Alternative Finance
(2)     Blockchain and Cryptocurrency
(3)     Challenger Banking
(4)     Crowdfunding
(5)     Data and Analytics
(6)     Islamic Enablers
(7)     P2P
(8)     Payment and Remittance and FX
(9)     Personal Finance Management, Trading, and Investment
(10)   Robo Advisers
(11)   TakaTech
(12)   Trading and Investment

Figure 1 highlights the deepening diversification within the industry. Last year, the largest segment combined crowdfunding and P2P companies, which together accounted for 31% of Islamic fintech companies. Today, they form two separate segments, each comprising approximately an equal number of industry representatives. In addition, the share of the payments segment has increased significantly. The growth rate of Islamic payment and remittance fintechs exceeds the growth rate of the industry as a whole.

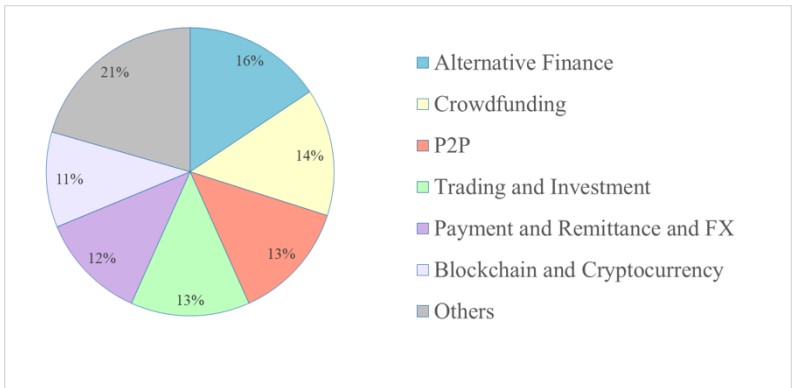

**Figure 1.** Segmentation of the Islamic fintech industry as of May 2021, % of companies. Source: own processing based on the statements of Global Islamic Economy Reports, DinarStandard.

Figure 2 provides a geographical breakdown of the six largest verticals. Among them, the blockchain and cryptocurrency segment appears to be the most diversified.

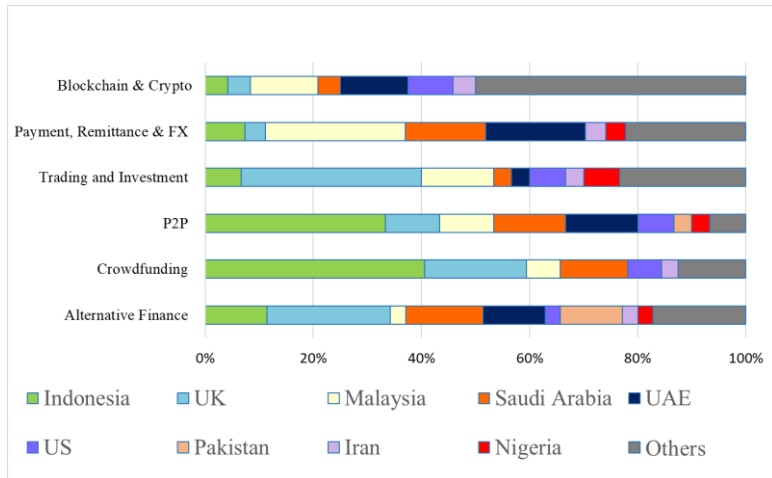

**Figure 2.** Geographical structure of the top six i-fintech verticals. Source: own processing based on the statements of Fintech Landscape (2021).

Within the segment, 50% of all i-fintechs come from the leading countries. As in the previous year, Indonesia, the United Kingdom, Malaysia, and the United Arab Emirates remain preferable jurisdictions for entrepreneurs from the five largest sectors.

In addition, a significant number of companies from Saudi Arabia should not go unnoticed. Indeed, the Saudi Arabian i-fintech industry has shown a dramatic increase from two companies in 2020 to 21 companies today.

The number of i-fintech companies can be seen to have risen across other countries. This tendency indicates the existence of unmet demand for Islamic fintech services. The contribution of each country to the overall increase in the number of companies is shown in Figure 3. Most of the new i-fintech entrants come from Indonesia, which has traditionally been considered a Muslim-majority country. Interestingly, the gain in the number of i-fintechs in Saudi Arabia equalled the gain in the United Kingdom.

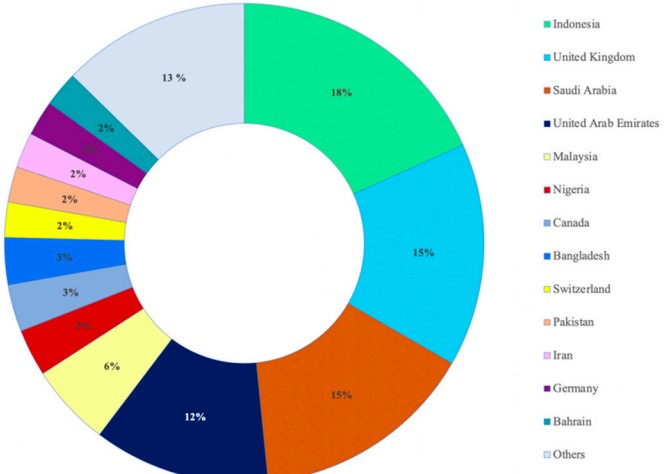

**Figure 3.** Islamic fintech industry annual growth, % of newcomers by country. Source: own processing based on the statements of FinTech Landscape, Finocracy.

The trends described above bring us to further investigation of the factors that determine the industry competitiveness of a country, i.e., the economy's potential and actual ability to enhance i-fintech industry scaling. The ability of an industry to scale in a given country depends on countless factors. Various approaches to the determination and measurement of these factors can be applied depending on the ultimate purpose. Considering the impossibility of measuring the i-fintech industry competitiveness directly, we assume

that it can be described within several dimensions. The key factors influencing i-fintech development include traditional Islamic financial institutions' capabilities, the development of information and computer technologies' infrastructure and facilities, the operational environment's vibrancy, outreach and awareness of Islamic finance, and the ethical aspect of finance.

In this regard, the indicator we constructed in (Glavina et al. 2020) has been revised. In our previous work, we introduced three levels of aggregation and used the min–max method to bring the initial indicators to one scale. Much of this paper has done likewise, but the calculation methods have been simplified and a wider range of variables has been arranged into a coherent structure.

The final indicator is based on data from the Islamic Fintech Landscape by IFN (Fintech Landscape 2021), the Islamic Finance Development Indicator by ICD Refinitiv (Zawya 2019), the Inclusive Internet Index by The Economist (The Inclusive Internet Index 2021), the Financial Access Survey (IMF 2021), The Global Competitiveness Report (WEF 2019), and World Bank studies (World Bank 2020a, 2020b). Moreover, the Global Islamic Fintech Index (GIFTI) of 64 countries was introduced by DinarStandard in 2021 (Global Islamic Fintech Report 2021), where 32 indicators were aggregated into 5 categories: Talent, Regulation, Infrastructure, Islamic Fintech Market and Ecosystem, and Capital. When combining the categories' indices into one value of a GIFTI at a country level, a heavier weighting was given to the Islamic Fintech Market and Ecosystem category. The methodology of the GIFTI's construction has much in common both with the indicator in our previous research and the one we have designed in the present paper.

Nevertheless, we suppose that there are several important dimensions uncovered within the GIFTI, such as the information environment or ethics dimensions of current Islamic markets. Considering these gaps, we have designed a composite indicator highlighting more aspects of the competitive environment of the Islamic fintech industry. A range of countries, either where the traditional Islamic financial industry was in place or that were OIC members, and 65 economies for which the necessary data were available were chosen for the final analysis.

The literature on Islamic fintech performance is becoming more and more popular, with many academic publications having been released in 2020 and 2021. There are a few timely books covering the effects and implications of fintech as well as its prospects for the Islamic financial industry. The book edited by Billah (2021, p. 465) provides an overview of how fintech can be adapted to Islamic principles and what are the threats and opportunities of Islamic fintech implementation. Hazik and Hassnian (2018, p. 236) explain the concepts of financial technologies and share insights into how fintech can enhance Islamic behaviour. The collection of articles edited by Alam and Nazim (2021, p. 257) evaluates the situation with Islamic fintech in the GCC region, analyses digital technologies from an Islamic perspective, and explores the future regulations of the segment (Alam, Nazim). Within this i-fintech literature, there are several variants of studies focused on certain aspects of Islamic fintech in certain countries. The study of Ali et al. (2019, pp. 73–108) analyses the impact of fintech on Islamic finance in Brunei and Malaysia and indicates how important it is for Islamic financial institutions to cope with the growth of fintech. Evans (2015, pp. 1–11) finds that blockchain technologies can conform with Sharia principles and are more appropriate, especially among the unbanked population and for small-scale cross-border trade. There are other studies representing a systematic literature review of the Islamic financial technologies area. Hasan et al. (2020, pp. 75–94) reviewed 16 studies focused on blockchain and cryptocurrency in Islamic fintech and showed that the growth of Islamic financial technologies will require the development of appropriate regulation and standards. Rabbani et al. collected 133 research studies related to Islamic fintech and proved the importance of appropriate regulatory framework evolvement (Rabbani and Khan 2020, pp. 65–86). There is another study by Hassan et al. (2020, pp. 93–116) that shows that Islamic fintech provides an equal playing field to Islamic finance to compete and grow and that in the post-COVID era i-fintech will play a significant role. Other researchers agree that the development

of Islamic fintech encourages sustainable development and fights poverty and hunger (Sahabuddin et al. 2019, pp. 651–56).

This article aspires to increase our knowledge about the competitiveness of the i-fintech segments of different countries. It focuses mainly on the designing of a composite indicator for cross-country analysis. Previous research has included fewer variables and key factors for designing the composite indicator in order to assess the competitive advantages of i-fintech sectors in different countries (Glavina et al. 2020). It may be assumed that the reconstructed indicator provides a more representative result. Since different issues of Islamic fintech have been addressed by many authors, including Vasenska et al. (2021), Evans (2015), Hazik and Hassnian (2018), Subagiyo (2019), Hassan et al. (2020), and Rabbani et al. (2021), this paper analyses the competitiveness of the Islamic fintech ecosystems at the national level based on the indicator developed by the authors.

In the outlined context, our article focuses on designing the original composite indicator of Islamic fintech development in order to assess the competitiveness of the countries in this field. After choosing the key factors of i-fintech development in different countries and introducing the composite indicator, the article aims at analysing economies with the highest scores and finding out their strengths and vulnerabilities.

The authors have chosen the following structure for the article. In the Introduction section, the fundamentals of the topic (development of Islamic finance, the importance of awareness, the role of Islamic fintech, and its development across segments and countries), along with the intentions and goals of the authors, are briefly outlined. The Materials and Methods section presents the measurement units and data sources for the variables of the composite indicator as well as the indicator itself and scores of the countries according to it. The Results section contains a detailed analysis of the development of the i-fintech sector in the top six countries. The Discussion and Conclusion sections are focused on the presentation of the most important findings compared with other studies and recommendations for further research.

## 2. Materials and Methods

To begin with, fintech is the result of finance and technology interpenetration. When it comes to Islamic fintech, the Sharia compliance aspect is also to be considered. Therefore, we assume that the more sophisticated Islamic finance and ICT sectors in a country are, the more favourable conditions for i-fintech development the country has. In addition, such a vibrant industry requires adequate regulation, as well as availability of funding and qualified personnel. To incorporate the ethos aspect of Islamic fintech, we further assume that the industry has more reasons to develop where there is a relevant business culture. Another essential pillar of Islamic fintech is knowledge. We expect the demand for Islamic fintech to depend on the level of public awareness of Islamic financial services. It is suggested that out of these assumptions a conceptual framework can be developed. These sub-indicators are presented in Figure 4.

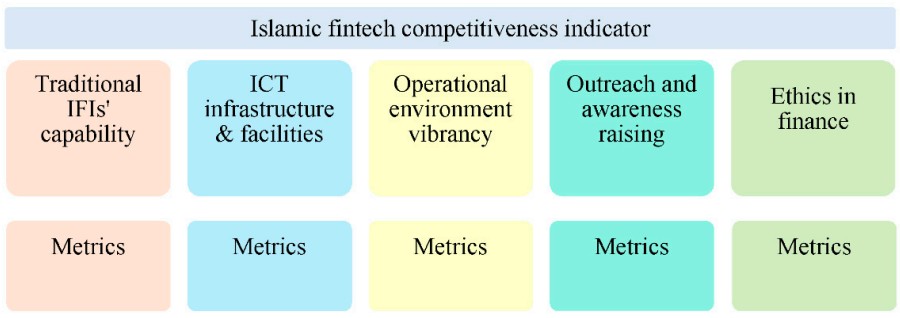

**Figure 4.** The structure of the Islamic fintech competitiveness indicator.

The sub-indicators that we have introduced may still be characterised only indirectly. To do this, we used the newest data currently available for 65 countries around the world.

The final country-level indicator was retrieved from 31 individual indicators included as explanatory variables in our model and denoted $X_1, \ldots, X_{31}$. To ensure comparability and the possibility of aggregation, we designed the metrics $\widetilde{X}_1, \ldots, \widetilde{X}_{31}$ and grouped them by sub-indicator. We have brought together the variables that are supposed to contribute to the concept of the corresponding dimensions, so the resulting multivariable structure of dimensions may be described as follows.

The first sub-indicator I1 aims at capturing the Traditional IFIs' capability of a country as a combination of metrics derived from Islamic banking assets (x1), Takaful/retakaful assets (x2), Other financial institutions assets (x3), Value of outstanding sukuk (x4), and Net asset value of Islamic funds (x5).

The ICT infrastructure and facilities sub-indicator I2 includes Fixed-line broadband subscribers (x6), Network coverage—min. 3G (x7), Internet users (x8), Smartphone cost (x9), Mobile subscribers (x10), and Secure Internet servers per 1 million people (x11).

Operational environment vibrancy (I3) seems to be a relatively heterogeneous sub-indicator because it combines Value of e-commerce (x12), Number of Islamic fintech companies (x13), Financing of SMEs (x14), Venture capital availability (x15), University–industry collaboration in R&D (x16), Government's responsiveness to change (x17), and Number of initiated regulatory sandboxes (x18).

At the next step, we defined the structure of the Outreach and awareness-raising dimension, which is measured through the sub-indicator I4 comprising such variables as Number of institutions offering training courses on Islamic finance (x19), Number of institutions offering degrees in Islamic finance (x20), Number of peer-reviewed/journal articles on Islamic finance (x21), Number of published research papers on Islamic finance (x22), Number of seminars (x23), Number of conferences (x24), and Number of exclusive and regional news articles (x25).

Finally, we added a group of variables measuring Ethics in finance to the sub-indicator denoted I5: Disclosed funds distributed to charity, zakat, and qard al hasan (x26), Average CSR disclosure index score (x27), Sharia governance regulations for Islamic finance institutions (x28), Centralised Sharia committee (x29), Number of scholars with SSB memberships (x30), and Disclosure index score (x31).

Most of the metrics can be obtained directly by normalising the initial values of the variables according to the formula:

$$\widetilde{X}(j) = \frac{X(j) - X_{min}(j)}{X_{max}(j) - X_{min}(j)} \times \frac{100}{n(j)} \tag{1}$$

where $\widetilde{X}(j)$ is the metric value included in the sub-indicator $Ij$, $j = 1, \ldots, 5$, $X(j)$ is the initial indicator value, $X_{max}(j)$ is the maximum value of the initial indicator, $X_{min}(j)$ is the minimum value of the initial indicator, and $n(j)$ is the overall number of metrics included in the sub-indicator $Ij$.

For several variables, we calculated the natural logarithm values prior to normalisation in order to reduce the impact of outliers. Table 1 provides details about the indicators used, including the sources of data as they are mentioned in references. The variables were marked with * if the corresponding metric was obtained by normalisation of the natural logarithm of the variable instead of its initial values. In this case, the metric was calculated using the formula:

$$\widetilde{X}(j) = \frac{lnX(j) - ln_{min}X(j)}{ln_{max}X(j) - ln_{min}X(j)} \times \frac{100}{n(j)}, \ X > 1 \tag{2}$$

**Table 1.** Measurement units and data sources of chosen variables.

| Variable | Name | Units | Ref. |
|---|---|---|---|
| **Indicator 1: Traditional IFIs' Capability** | | | |
| x1 * | Islamic banking assets | USD | Islamic Finance Development Indicator ([Zawya 2019](#) ) |
| x2 * | Takaful/retakaful assets | USD | |
| x3 * | Other financial institutions assets | USD | |
| x4 * | Value of outstanding sukuk | USD | |
| x5 * | Net asset value of Islamic funds | USD | |
| **Indicator 2: ICT Infrastructure and Facilities** | | | |
| x6 | Fixed-line broadband subscribers | Number per 100 inhabitants | The Inclusive Internet Index ([The Inclusive Internet Index 2021](#)) |
| x7 | Network coverage—min. 3G | % of population | |
| x8 | Internet users | % of households | |
| x9 | Smartphone cost | Score of 0–100; 100 = most affordable | |
| x10 | Mobile subscribers | Number per 100 inhabitants | |
| x11 * | Secure Internet servers per 1 million people | Number per 1 million people | Financial Access Survey ([IMF 2021](#)) |
| **Indicator 3: Operational Environment Vibrancy** | | | |
| x12 | Value of e-commerce | % of responses | The Inclusive Internet Index ([The Inclusive Internet Index 2021](#)) |
| x13 | Number of Islamic fintech companies | Number | Islamic Fintech Landscape ([Fintech Landscape 2021](#)) |
| x14 | Financing of SMEs | 1–7 Score | The Global Competitiveness Report ([WEF 2019](#)) |
| x15 | Venture capital availability | 1–7 Score | |
| x16 | University–industry collaboration in R&D | 1–7 Score | |
| x17 | Government's responsiveness to change | 1–7 Score | |
| x18 | Number of initiated regulatory sandboxes | Number | Key Data from Regulatory Sandboxes across the Globe ([World Bank 2020a](#)) |
| **Indicator 4: Outreach and Awareness-Raising** | | | |
| x19 * | Number of institutions offering training courses on Islamic finance | Number | Islamic Finance Development Indicator ([Zawya 2019](#)) |
| x20 * | Number of institutions offering degrees in Islamic finance | Number | |
| x21* | Number of peer-reviewed/journal articles on Islamic finance | Number | |
| x22 * | Number of published research papers on Islamic finance | Number | |
| x23 * | Number of seminars | Number | |
| x24 * | Number of conferences | Number | |
| x25 * | Number of exclusive and regional news articles | Number | |

**Table 1.** *Cont.*

| Variable | Name | Units | Ref. |
|---|---|---|---|
| | **Indicator 5: Ethics in Finance** | | |
| x26 | Disclosed funds distributed to charity, zakat, and qard al hasan | USD million | |
| x27 | Average CSR disclosure index score | Score | Islamic Finance Development Indicator (Zawya 2019) |
| x28 | Sharia governance regulations for Islamic finance institutions | 0/1 | |
| x29 | Centralised Sharia committee | 0/1 | |
| x30 | Number of scholars with SSB memberships | Number | |
| x31 | Disclosure index score | Score | |

* Explained on page 7.

In order to make the interpretation of our indicator easier despite the complexity of the phenomenon, each sub-indicator was calculated by Formula (3). The highest possible value of each sub-indicator is 100 and this can be obtained only if all of its metrics have the maximum values at once.

$$I_j = \sum \widetilde{X}(j) \tag{3}$$

where $I_j$ is the $j$th sub-indicator and $\widetilde{X}(j)$ stands for a metric included in the $I_j$ sub-indicator.

The country-level indicator of the i-fintech competitiveness is based on five sub-indicators and is calculated as the average value:

$$I_{IFTC} = \frac{\sum_{j=1}^{5} I_j}{5} \tag{4}$$

Finally, we have ranked the countries in descending order of the i-fintech competitiveness indicator. The estimation results both for the indicator and the sub-indicators by country are presented in Table 2.

**Table 2.** Countries by IIFTC score and scores of sub-indicators.

| № | Economy | i-Fintech Competitiveness Indicator Score | I1 | I2 | I3 | I4 | I5 |
|---|---|---|---|---|---|---|---|
| 1 | Malaysia | 76.01 | 98.94 | 57.97 | 69.62 | 87.76 | 65.76 |
| 2 | UAE | 73.47 | 91.94 | 78.42 | 72.48 | 63.9 | 60.58 |
| 3 | Indonesia | 69.92 | 90.37 | 48.26 | 62.99 | 82.16 | 65.82 |
| 4 | Saudi Arabia | 62.15 | 98.08 | 57.94 | 61.23 | 58.69 | 34.82 |
| 5 | Bahrain | 60.67 | 82 | 60.66 | 48.25 | 47.47 | 64.97 |
| 6 | United Kingdom | 57.64 | 52.04 | 80.88 | 72.65 | 63.68 | 18.93 |
| 7 | Qatar | 57.16 | 89.61 | 66.96 | 57.08 | 36.17 | 35.99 |
| 8 | Pakistan | 56.87 | 85.52 | 27.4 | 39.86 | 72.93 | 58.66 |
| 9 | Oman | 53.57 | 65.51 | 59.54 | 43.23 | 29.7 | 69.89 |
| 10 | Kuwait | 52.52 | 87.89 | 63.23 | 40.79 | 29.15 | 41.57 |

**Table 2.** *Cont.*

| № | Economy | i-Fintech Competitiveness Indicator Score | I1 | I2 | I3 | I4 | I5 |
|---|---------|-------------------------------------------|-----|------|------|------|------|
| 11 | USA | 52.07 | 47.39 | 81.76 | 84.65 | 43.82 | 2.74 |
| 12 | Singapore | 49.61 | 47.61 | 82.32 | 70.81 | 18.88 | 28.42 |
| 13 | Jordan | 46.4 | 76.69 | 38 | 40.97 | 31.59 | 44.76 |
| 14 | Turkey | 46.05 | 67.5 | 58.08 | 34.84 | 47.02 | 22.81 |
| 15 | Bangladesh | 41.89 | 66.38 | 37.82 | 20.71 | 30.05 | 54.49 |
| 16 | Hong Kong SAR | 37.25 | 29.42 | 90.34 | 63.17 | 3.3 | 0 |
| 17 | Nigeria | 36.78 | 42.49 | 25.47 | 20.74 | 36.95 | 58.28 |
| 18 | Switzerland | 36.65 | 18.32 | 83.24 | 60.13 | 8.14 | 13.43 |
| 19 | Australia | 34.32 | 14.1 | 78.53 | 43.41 | 25.93 | 9.63 |
| 20 | Sri Lanka | 32.77 | 45.11 | 40.37 | 30.66 | 20.42 | 27.28 |
| 21 | India | 31.83 | 28.43 | 40 | 53.69 | 27.96 | 9.07 |
| 22 | Egypt | 31.75 | 48.27 | 43.32 | 33.49 | 21.38 | 12.3 |
| 23 | Canada | 31.57 | 13.21 | 76.88 | 49.22 | 13.81 | 4.72 |
| 24 | Iran | 31.25 | 59.5 | 48.59 | 17.65 | 30.2 | 0.31 |
| 25 | Thailand | 30.82 | 30.34 | 60.38 | 57.4 | 5.57 | 0.43 |
| 26 | France | 29.92 | 7.13 | 78.6 | 44.61 | 18.73 | 0.52 |
| 27 | Morocco | 29.73 | 28.17 | 52.33 | 24.53 | 26.97 | 16.67 |
| 28 | Tunisia | 29.64 | 41.39 | 45.36 | 16.46 | 27.26 | 17.71 |
| 29 | Kazakhstan | 29.26 | 27.56 | 58.88 | 29.2 | 12.35 | 18.3 |
| 30 | Germany | 28.97 | 0 | 78.03 | 56.42 | 9.7 | 0.69 |
| 31 | Netherlands | 28.94 | 0 | 81.57 | 55.53 | 7.6 | 0 |
| 32 | Japan | 28.15 | 0 | 78.86 | 51.96 | 9.95 | 0 |
| 33 | Kenya | 27.86 | 26.43 | 32.53 | 38.89 | 29.98 | 11.47 |
| 34 | Denmark | 27.43 | 0 | 80.71 | 51.58 | 4.84 | 0 |
| 35 | Ireland | 27.27 | 16.08 | 76.2 | 43.49 | 0.57 | 0 |
| 36 | New Zealand | 26.39 | 0 | 77.99 | 47.09 | 6.62 | 0.26 |
| 37 | Sweden | 26.31 | 0 | 77.56 | 50.52 | 3.46 | 0 |
| 38 | South Korea | 25.74 | 0 | 76.37 | 46.83 | 5.49 | 0 |
| 39 | China | 24.97 | 0 | 61.11 | 51.48 | 12.25 | 0 |
| 40 | Azerbaijan | 24.73 | 13.56 | 55.4 | 51.3 | 3.41 | 0 |
| 41 | Spain | 24.58 | 0 | 74.43 | 36.73 | 11.72 | 0 |
| 42 | Belgium | 24.02 | 0 | 79.53 | 38.82 | 1.76 | 0 |
| 43 | Philippines | 23.52 | 12.11 | 38.85 | 40.87 | 9.11 | 16.67 |
| 44 | Lebanon | 23.44 | 14.94 | 48.44 | 21.44 | 14.27 | 18.14 |
| 45 | Russia | 22.04 | 0 | 65.83 | 30.4 | 13.96 | 0 |
| 46 | Vietnam | 19.88 | 0 | 50.08 | 34.9 | 2.98 | 11.45 |
| 47 | Algeria | 19.34 | 16.23 | 44.39 | 24.95 | 10.17 | 0.95 |
| 48 | Trinidad and Tobago | 17.91 | 13.01 | 61.52 | 14.88 | 0.04 | 0.09 |
| 49 | Brazil | 17.11 | 0 | 54.34 | 28.78 | 2.42 | 0 |

**Table 2.** *Cont.*

| № | Economy | i-Fintech Competitiveness Indicator Score | I1 | I2 | I3 | I4 | I5 |
|---|---------|---------|------|-------|-------|------|-------|
| 50 | Senegal | 17.02 | 31.72 | 27.9 | 24.26 | 0.93 | 0.26 |
| 51 | Cambodia | 15.72 | 0 | 36.2 | 30.3 | 0.66 | 11.45 |
| 52 | Tanzania | 15.63 | 13.68 | 26.72 | 32.43 | 3.91 | 1.44 |
| 53 | Botswana | 14.89 | 0 | 51.85 | 21.85 | 0.75 | 0 |
| 54 | Rwanda | 13.45 | 0 | 25.87 | 40.64 | 0.75 | 0 |
| 55 | Ghana | 12.71 | 0 | 34.53 | 26.29 | 2.36 | 0.35 |
| 56 | Ethiopia | 12.42 | 13.84 | 15.21 | 22.69 | 0.81 | 9.54 |
| 57 | Guinea | 11.27 | 0 | 11.01 | 44.23 | 0.84 | 0.26 |
| 58 | Mali | 11.13 | 14.91 | 18.81 | 21.06 | 0.87 | 0 |
| 59 | Zambia | 10.84 | 0 | 24.82 | 11.69 | 0.75 | 16.93 |
| 60 | Uganda | 9.34 | 0 | 17.62 | 25.42 | 3.67 | 0 |
| 61 | Gabon | 9.09 | 0 | 38.41 | 6.28 | 0.75 | 0 |
| 62 | Cameroon | 9.01 | 0 | 24.7 | 19.31 | 1.05 | 0 |
| 63 | Benin | 7.55 | 0 | 17.15 | 19.82 | 0.75 | 0 |
| 64 | Mozambique | 6.5 | 0 | 17.97 | 13.8 | 0.75 | 0 |
| 65 | Burkina Faso | 4.57 | 0 | 9.57 | 12.13 | 1.13 | 0 |

## 3. Results

Before we move on to the analysis of the assessment results, let us emphasise that the indicator is calculated to assess the comparative degree of the actual and potential development of the Islamic fintech sector in a given country. The structure of the composite indicator takes into account a limited number of factors; moreover, the model specification does not reflect their actual relationships because it was chosen to ensure easy interpretation of the final results. In addition, data for a number of indicators were updated and published a year ago or earlier. Given the current state of the global economy, the inclusion of new data will lead to significant changes in the final indicators. However, comparing countries within the indicator and sub-indicators allows us to determine the relative level of industry readiness right now, providing the necessary foundation for further refinement of the indicator.

In terms of the considered aspects of the modern i-fintech industry, Malaysia ranks first (a score of 76.01) in the overall ranking and therefore is regarded as the most competitive region. Malaysia leads the Traditional IFIs' capability dimension with a near-perfect score of 98.94, while its lowest score of 57.97 indicates the vulnerability of the ICT infrastructure and facilities component. Malaysia also tops the Outreach and awareness-raising pillar and features in the top five in the Operational environment vibrancy and Ethics in finance pillars. We expect the further assessment of the ICT aspect of Malaysia's i-fintech competitiveness to be higher, even though the economy is suffering from the difficulties caused by the COVID-19 pandemic. It is ICT infrastructure and effective communication between business and government that are the subjects of systematic optimisation in Malaysia. The roadmap to the desired level of performance has been set out in the Malaysian Economy Digitalisation Program, which was published by Malaysia Digital Economy Corporation (MDEC) this year (Malaysia Digital Economy Blueprint 2021). With the consequences of the COVID-19 pandemic already taken into account, the implementation of the strategy implies a projected increase in the digital economy's contribution to GDP up to 22.6%. E-commerce adoption is also supposed to be accomplished by 875,000 small and medium-sized businesses, and digitalisation is expected to cover all national economy participant groups. In addition, Malaysia's geographic location may become an additional competi-

tive advantage: Southeast Asia is the fourth-largest Internet market, where the region's cumulative e-commerce revenues estimates exceeded USD25 billion as at the end of 2020.

The United Arab Emirates ranks second (a score of 82.6) globally, performing noticeably better than Malaysia in the ICT infrastructure and facilities dimension (a score of 78.42) and therefore ranking 12th within it. The United Arab Emirates features in the top five in the four remaining pillars. The lowest score of 60.58 in the Ethics in finance pillar is mainly the result of a comparatively low disclosure score. In fact, 60 points can hardly be referred to as a low score, but still the average level of expertise among SSB members in the United Arab Emirates and several other countries with the same model of Sharia regulation is lower than in countries with other models. At the same time, the regulatory system of the United Arab Emirates does not prohibit cross-membership, which means that a scholar can hold positions in several SSBs at once. Given the scarcity of qualified Islamic scholars, this may lead to ineffective and overpriced expert services. On the other hand, the same study emphasises the value of the qualified scholars' reputation and the general lack of Islamic scholars with expertise in both Sharia law and finance; thus, it cannot be said with certainty that the UAE's Sharia regulatory system requires any improvements.

Ranked as the third most competitive economy (a score of 69.92), Indonesia features in the top five in the Traditional IFIs' capability, Outreach and awareness-raising, and Ethics in finance pillars, and in terms of the Operational environment vibrancy pillar the economy is ranked seventh. However, Indonesia struggles on the ICT sub-indicator (a score of 48.26), in which the country is more than halfway to the frontier. Despite its noticeably wide market, we expect a decrease in GDP to be reported and further contributions to the ICT industry to be hindered. According to McKinsey, approximately 60% of Indonesia's GDP formation is being provided by micro-, small-, and medium-sized enterprises, which also offer 97% of the jobs in the country (Agarwal Rajat et al. 2021). The total number of such enterprises is estimated to reach 63 million, which means that Indonesia's economy and finance may take more time to recover and rebuild. In this case, we witness a need for targeted measures. Facilitation of the country's ICT infrastructure implies large expenditures on tangible components, while there is less concern related to the digitalisation of businesses, households, and the government. The return of the country to the pre-crisis level of economic growth, on the one hand, will become easier with the introduction of digital technologies, but, on the other hand, the path of the country to Industry 4.0 implementation is complicated because of the blow to the real sector.

Saudi Arabia ranks as the fourth most (a score of 62.15) competitive jurisdiction ready to scale i-fintech. It ranks second behind Malaysia in the Traditional IFIs' capability dimension, and the gap between the two economies is less than 1 point. In the Outreach and awareness-raising dimension and in the Operational environment vibrancy dimension, Saudi Arabia ranks among the top 10 countries, and in the Ethics in finance dimension it ranks among the top 15 countries (12th), despite the quite low score (34.82) in the latter. Again, in the ICT infrastructure and facilities dimension the economy lags far behind the leaders and holds the 31st position within the sub-indicator with a score of 57.94, thus remaining above the average. We have already mentioned the extremely high proportion of newcomers evidenced within the Saudi Arabian i-fintech market. This may arise from an equally favourable operating environment, the development of the information field (the Outreach and awareness-raising dimension), and the provision of the economy with ICT infrastructure. According to KPMG (KPMG 2020), the interaction across all stakeholders is in the country's current focus; thus, its impact is expected to some extent within all four sub-indicators. Moreover, Islamic finance industry incumbents in Saudi Arabia contribute significantly to the demand for i-fintech instead of competing with one another. The Saudi Arabia Financial Sector Development Agenda lists initiatives that were designed to provide crucial prerequisites for enhancing the development of the industry. We suppose the inflow of newcomers to the i-fintech in the country to be a return on the efforts made in terms of the projects aimed at the availability of financing for small- and medium-sized businesses,

or providing the economy with an electronic payment system and related services, or the introduction of technologies such as eKYC.

Bahrain ranks fifth overall with a score of 60.67 and can rely on a highly developed Islamic financial sector (a Traditional IFIs' capability dimension score of 82.00) while addressing its vulnerability that may arise as an issue from the Operational environment vibrancy pillar or the Outreach and awareness-raising one. However, for this individual case, one should be careful about the estimates obtained, since the economy of Bahrain is several times smaller than that of the other countries in our study. Indeed, Bahrain is believed to have been one of the main hubs of Islamic finance for years. It has an established and effective Shariah regulatory system, a high concentration of Islamic financial sector assets, and an intensive technology support policy in the financial sector.

One of the best performing i-fintech enabling economies is the United Kingdom, which ranks sixth overall with a total score of 57.64. The United Kingdom stands out in its capacity as the only European country with an advanced level of Islamic financial system development. The competitiveness of the i-fintech industry is higher here than in many Muslim countries. The country appears in the top half of the rankings in all considered dimensions except for the Ethics in finance dimension, where it ranks 16th with a score of 18.93. Despite the halal culture in finance being relatively underdeveloped, the United Kingdom remains one of the most attractive jurisdictions for new Islamic fintech companies. The existence of a sufficiently sophisticated national culture of halal finance may affect the Islamic fintech sector's ability to find consumers. Apparently, the absence of such a culture might be compensated for by the strong performance across the Islamic financial sector, especially while the U.K.'s halal media landscape (the Outreach and awareness-raising dimension) is fueling public interest in the Islamic financial model. Given the level of ICT development in the United Kingdom, the process of capturing the attention of the target audiences is being simplified and accelerated, allowing businesses to find and activate the potential effective demand. It is clear that in Muslim-majority countries it is even more crucial to increase public awareness of Islamic finance, as well as to deepen and expand the information technology field. If a lack of experienced agents in the Islamic financial niche or ICT is being observed, while awareness remains high, the case might be regarded as a signal for new entrants to take advantage of existing market opportunities.

## 4. Discussion

We introduced the competitiveness of Islamic fintech in a particular country as a combination of five fundamental components: Traditional Islamic financial institutions' capacity (I1), ICT infrastructure and facilities (I2), Operational environment vibrancy (I3), Outreach and awareness-raising (I4), and Ethics in finance (I5). These aspects, as well as the variables included in them, are not exhaustive for characterising the development of the industry. On the one hand, due to the fact that all variables are weighted equally in the final indicator, the resulting indicators are easy to interpret. On the other hand, we may be downplaying the importance of factors such as ICT and exaggerating the importance of Ethics in finance.

Despite the differences in the methodology for compiling our indicator and the global Islamic Fintech (GIFT) Index, the final rankings of the countries have much in common. For example, nine out of the ten most competitive countries, according to our version, were also included in the top 10 according to the GIFT Index, the FinTech Adoption Index, and the Islamic finance development indicator. Nevertheless, both models have significant potential for improvement.

First of all, it should be noted that the composite indicator compiled by us includes the aspects that we considered important and they do not provide the full picture. It should also be noted that during the research process we faced a lack of data for each particular period; therefore, we used the most recent data available. We have tried to keep our model as objective and easy to interpret as possible. However, this also imposes certain limitations: the model leaves out possible differences in the importance of the factors included. In

addition, we looked at countries with widely varying characteristics, such as population and GDP. In the future, attention to regional characteristics based on such variables will qualitatively improve the model.

Since the COVID-19 pandemic appeared to be a catalyst for the widespread adoption of fintech, further research is needed to monitor intra-industry trends and their role in moving towards sustainable development. Time will also allow us to determine if the factors and their weights have been chosen correctly. Therefore, future work in this direction involves improving the structure of factors, the basic model of the indicator, data actualisation, and expansion of the geographical and temporal coverage of the study.

Moreover, deeper analysis of the dimensions that contribute to the competitiveness of the i-fintech industry in an economy allowed us to define the areas of concern across the leading economies. All of them can be addressed through the framework of state development programs, unless more severe consequences of the COVID-19 pandemic arise.

On the one hand, the COVID-19 pandemic has caused a multidimensional aggregate demand and supply crisis, but, on the other hand, it appears to have been a catalyst for the widespread adoption of fintech and the growing appetite for Islamic finance globally. During the last year, the Islamic fintech industry has shown its resilience by growing further worldwide; thus, we expect more studies in the near future, especially those investigating i-fintech's role in addressing the adverse effects of the COVID-19 pandemic.

Considering the evaluation of and comparisons between economies, research is needed to monitor intra-industry trends and their role in moving towards sustainable development. Future work in this direction involves improving the structure of factors, the basic model of the indicator, data actualisation, and expansion of the geographical and temporal coverage of the study.

## 5. Conclusions

The Islamic financial industry is demonstrating steady development and becoming more diversified. Islamic fintech is one of the important growth drivers. The COVID-19 pandemic boosted i-fintech projects and led to a better awareness of Islamic finance. The results of this study also suggest that Islamic finance education supported by the increased public awareness may unlock additional efficiency of meeting supply and demand within the market. The offering of specific and applied courses and programs in Islamic finance by universities and Islamic financial organisations and regulators, particularly in Islamic financial technologies, will contribute to further awareness and development of Islamic financial business and i-fintech.

The rise of Islamic fintech, like that of conventional financial technologies, provides financial inclusion for the world's unbanked population, facilitates access to Islamic financial products, widens opportunities for ethical investments, including ESG-rich opportunities, leads to poverty reductions, and promotes gender equality, which is extremely important for developing and emerging economies.

i-Fintech is becoming more diversified and geographically spread, although about half of the businesses are concentrated in the leading countries—Malaysia, the United Kingdom, and the United Arab Emirates.

We assume that the key factors affecting i-fintech are traditional Islamic financial institutions' capabilities, the development of infrastructure and facilities for ICT, the operational environment's vibrancy, the outreach and awareness of Islamic finance, and the ethical aspect of finance.

We obtained estimated values of these five dimensions of the Islamic fintech industry's competitiveness and then combined them into a single composite indicator at a country level.

The evaluation results indicate that the six most favourable jurisdictions for Islamic fintechs are Malaysia (76.01), Indonesia (73.47), the United Arab Emirates (69.92), Saudi

Arabia (62.15), Bahrain (60.67), and Great Britain (57.64). Among the top ten countries, nine are OIC members.

This composite indicator provides a competitiveness assessment of the countries. Analysis of the strengths and advantages can help businesses select an appropriate jurisdiction. At the same time, our results shed light on the weaknesses and vulnerabilities of the Islamic fintech sector's development, which may offer guidance to regulators in addressing challenges around the development of i-fintech in their countries.

**Author Contributions:** Conceptualization, S.G. and A.T.; methodology, S.G. and A.T.; software, A.T.; validation, S.G. and A.T.; formal analysis, A.T. and S.G.; investigation, S.G.; resources, I.A.; data curation, S.G.; writing—original draft preparation, A.T.; writing—review and editing, I.A.; visualization, A.T.; supervision, S.G.; project administration, I.A.; funding acquisition, I.A. All authors have read and agreed to the published version of the manuscript.

**Funding:** This article was prepared within the framework of Research Project No. 203367-0-000 "Global transformation of world finance" (RUDN University, 2021).

**Data Availability Statement:** Not applicable.

**Conflicts of Interest:** The authors declare no conflict of interest.

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
