# Peer review of "Assessment of the Competitiveness of Islamic Fintech Implementation: A Composite Indicator for Cross-Country Analysis"

_jrfm, doi:10.3390/jrfm14120602_

Round 1

Reviewer 1 Report

The authors of the paper Assessment of Competitiveness of Islamic Fintech Implementa-tion: Composite Indicator for Cross-Country Analysis, present a relevant topic, namely the presentation of an "original composite indicator of the competitiveness of Islamic fintech adoption" aspects particularly relevant regionally and globally in the context of implementation "Islamic financial technologies in different countries around the world", the work can have a multiplier effect due to the results obtained.

Concepts, bibliographic sources and citations are appropriately mentioned in the paper. For example, the authors mention and justify through bibliographic sources of different authors, scientific arguments through works such as fintech became the focus of authors' studies (Glavina, Aidrus et. El. 2020) and others only last year (Rabbani, Khan 2020).

The research methodology is appropriate and directly oriented to the topic of the problem analyzed by the authors, respectively the use of indicators analysis methodologies to support the analysis of the scenario, such as the interpretation of variables that should contribute to the concept of appropriate dimensions, so that the resulting multivariable structure dimensions and described in the paper.

The results of the research are presented by the authors of the research, respectively they reflect and interpret the optimal solutions on “international financial interaction and therefore the globalization of Islamic finance required supranational regulators, including, for example, accounting and Audit Organization for financial institutions (AAOIFI) and the Islamic Financial Services Council (IFSB). ”. The results are presented more from an applicative point of view, which is why we suggest the authors of the research to present in a separate paragraph the personal scientific contributions to the literature.

The conclusions are presented in the discussion chapter, so we suggest that the authors present both the discussions and the conclusions in two separate paragraphs, especially since the authors "obtained estimated values of the five dimensions of the competitiveness of the Islamic fintech industry" and " a single composite indicator at country level”. However, we suggest to the authors of the research to highlight as we mentioned in the chapter the results and personal scientific contributions. Moreover, we suggest presenting both the limitations of the study, future research is adequately presented by the authors of the research in the study.

We congratulate the research team, and after reviewing the paper, we propose for acceptance the paper.

Author Response

Dear reviewer,

Thank you for your revision and recommendations. We followed all of them.

We presented a paragraph in the Introduction where we talk about our findings and contribution in the literature.

We discussed the limitations of our study and our future researches too.

We presented discussion and conclusions separately. 

Thank you and kind regards,

Reviewer 2 Report

Dear authors,

you provide valuable scientific manuscript.

My recommendation is linked to the references. Extend your literature review and extend Discussion (compare your results to similar studies).

For example:

Musa, H., Musova, Z., Natorin, V., Lazaroiu, G., & Martin Boda, M. (2021). Comparison of factors influencing liquidity of European Islamic and conventional banks. Oeconomia Copernicana12(2), 375–398.

Musa, H., Natorin, V., Musova, Z., & Durana, P. (2020). Comparison of the efficiency measurement of the conventional and Islamic banks. Oeconomia Copernicana11(1), 29–58.

Vasenska, I., Dimitrov, P., Koyundzhiyska-Davidkova, B., Krastev, V., Durana, P., Poulaki, I. Financial Transactions Using FINTECH during the Covid-19 Crisis in Bulgaria. Risks 2021, 9, 48.

Good luck in your future work.

Author Response

Dear Reviewer,

Thank you for your revision and recommendations. We followed all your recommendations.

We extended the literature review and discussion. We added Conclusion sector.

We analyzed the articles you sent and found them very useful, thus we included them in our research.

Thank you and kidn regards

Reviewer 3 Report

Please develop the abstract of the paper.

Please supplement and restructure the Introduction section to include:

  1. One paragraph that positions the paper compared to the academic literature.
  2. One paragraph that details the contribution of the paper to the literature.
  3. One paragraph that summarizes/presents the results of the study. This paragraph should be the ultimate paragraph in the introduction before the presentation of the remaining parts of the chapter.
  4. The last paragraph of the introduction should “introduce” the remaining parts of the paper.

Conclusions need to be extended to reflect the findings of the study and should be extended to discussion with literature.

Keeping in mind the points above, I suggest the revision for the
paper.

Author Response

Dear Reviewer,

Thank you for you revision. We followoed all your valuable recommendations.

We developed the abstract of the paper.

We extended and restructured Introduction:

  • we added one paragraph that positions the paper compared to the academic literature
  • we wrote about our contribution to the literature
  • we prepared one paragraph that presents the results of the sudy
  • we presented the remaining parts of the paper in the last paragraph

We added Conclusion and extended Discussion. As there is no previous studies with the focus on competitive assessments and attempts to develop an index, so we couldn't refer to the literature in Discussion.

Thank you and kind regards,

Round 2

Reviewer 2 Report

Dear authors,

thank you for your effort. I recommend publishing in present form.

Reviewer 3 Report

I recommend the article for publication in its present form.